# Hard Carbons for Use as Electrodes in Li-S and Li-ion Batteries

**DOI:** 10.3390/nano12081349

**Published:** 2022-04-14

**Authors:** Alfonso Pozio, Mariasole Di Carli, Annalisa Aurora, Mauro Falconieri, Livia Della Seta, Pier Paolo Prosini

**Affiliations:** 1TERIN-PSU-ABI, ENEA, C.R. Casaccia, Via Anguillarese 301, S. Maria di Galeria, 00123 Rome, Italy; alfonso.pozio@enea.it (A.P.); mariasole.dicarli@enea.it (M.D.C.); annalisa.aurora@enea.it (A.A.); livia.dellaseta@enea.it (L.D.S.); 2FSN-TECFIS, ENEA, C.R. Casaccia, Via Anguillarese 301, S. Maria di Galeria, 00123 Rome, Italy; mauro.falconieri@enea.it

**Keywords:** activated carbon, waste biomass, pyrolysis, Li-sulfur battery, Li-ion battery

## Abstract

Activated hard carbons, obtained from the pyrolysis of various waste biomasses, were prepared and characterized for use as the active material for the fabrication of battery electrodes. The preparation consisted of a pyrolysis process, followed by an activation with KOH and a further high-temperature thermal process. TG and DTA were used to discriminate the steps of the activation process, while SEM, XRD, and Raman characterization were employed to evaluate the effects of activation. The activated carbons were tested as electrodes in lithium-sulfur and lithium-ion batteries. The carbonaceous materials coming from cherry stones and walnut shells have proved to be particularly suitable as electrode components. When used as anodes in lithium-ion batteries, both carbons exhibited a high first cycle discharge capacity, which was not restored during the next charge. After the first two cycles, in which there was a marked loss of capacity, both electrodes showed good reversibility. When used as cathodes in lithium-sulfur batteries, both carbons exhibited good catalytic activity against the redox reaction involving sulfur species with good cycle stability and satisfactory Coulombic efficiency.

## 1. Introduction

Lithium-ion batteries (LIB) represent one of the best performing energy storage systems, in terms of cycle life and energy density. However, the high costs and environmental problems of some components used for their realization represent a serious disadvantage, particularly for large-scale application [1]. Recently, alternative technologies to LIB have been developed. These technologies could lead to significant advantages in terms of costs, sustainability, or environmental impact. Among these alternative technologies, lithium-sulfur (Li-S) batteries are currently the subject of intense experimentation. Regrettably, the use of a metallic Li anode makes the cell manufacturing process dangerous and increases the production cost. Furthermore, it introduces drawbacks during operation, such as the pulverization/formation of “mossy Li”, the formation of Li dendrites, or the decomposition of the electrolyte, which gradually lead to cell degradation. To overcome these problems, alternatives to lithium metal, such as LIB, have been proposed. In this case, the sulfur must be replaced with lithium sulfide: graphite-Li_2_S [2,3,4,5], metal oxide-Li_2_S [6], silicon (Si)-Li_2_S [7,8], and phosphorus-Li_2_S [9] have been successfully explored. Research activities are mainly aimed at improving the performance of active materials in terms of electrochemical characteristics, cost reduction, and lower environmental impact. As regards this last aspect, intense R&D studies have aimed to develop carbonaceous materials and graphitic or amorphous carbons, obtained by thermal treatment of biomass. Recently, porous carbonaceous materials have been widely reported as promising Li-S battery electrodes, able to promote the kinetics of the sulfur conversion reaction, tolerate volume expansion in the sulfur/sulfide transformation, and suppress the polysulfide shuttling process [10]. Two main features make the carbons noteworthy to be used as battery electrodes: a high electrical conductivity, which guarantees a rapid transfer of electrons [11], and a proper micro/mesoporosity ratio, which allows the sulfur to be hosted inside the structure without sequestering it irreversibly [12]. Therefore, to be used as an electrode in Li-S batteries, an ideal carbonaceous material should be capable of ensuring good conductivity and high capacity. To this end, it is of great importance to design porous carbons with hierarchical distributions of pore sizes. Among the numerous synthesis routes, biomass carbonization represents an economic strategy for preparing porous carbon on a large scale. However, these materials have irregular morphologies/nanostructures that do not allow for rapid electron/ion transfer. Therefore, the carbon architecture tailoring could represent a promising way to prepare a multifunctional carbon matrix for high-performance Li-S batteries. Various synthesis methods have been developed to achieve the desired architectures, including the use of templates, the combination of carbonization and activation processes, and the synthesis pathway using molten salts. This last technique allows to obtain carbons with high specific surfaces and ordered pore size distributions [13]. Activation of carbon by hydroxides represents a well-known technique that continues to be intensely pursued for the unique properties of the resulting activated carbon. In particular, the synthesis based on the chemical reaction between the carbon precursor and the alkaline hydroxide offers the best reaction products in terms of high porosity and uniform pore size distributions. These two properties are important for applications such as storage ions and make this carbon suitable for the construction of accumulators of new generation [14]. Compared to well-ordered graphite, disordered carbons exhibit a wide variable property, with an increased level of structural disorder in the stacking of sp^2^-hybridized covalent lattice sheets. Generally, disordered carbons are classified into two categories, (i) soft carbons, which are graphitizable at high temperatures (>2500 °C) and (ii) hard carbons, which cannot be graphitized and, through high-temperature treatment (3000 °C), give rise to glassy carbon [15]. This difference arises from the choice of precursor, as hydrogen-containing thermoplastic precursors (e.g., hydrocarbons, polyvinyl chloride, pitch, benzene) lead to soft carbons, while thermosetting precursors (e.g., sugars, phenol resins, biomass) lead to hard carbons. During pyrolysis, the release of heteroatoms (e.g., H, O, N) as volatile compounds (e.g., CH_4_, CO_2_, H_2_O, CO, NO_x_, etc.) decreases the overall carbon content, with a significant loss of mass typically observed between 250 and 500 °C. To improve the electrochemical performance of the materials, an additional heat treatment was carried out after impregnation with KOH. Although KOH activation is a common method to generate extended porosity in carbons [16], the activation mechanisms are not clear due to the high number of experimental variables and the diverse reactivity of the precursors [17]. Several analysis techniques, such as SEM analysis, X-ray diffraction (XRD), thermic gravimetric and differential thermal analysis (TG-DTA), and Raman spectroscopy, have been employed to analyze hard carbons [18,19,20]. In this paper, the synthesis of KOH-activated carbon coming different biomass sources was carried out. The carbons were analyzed to evaluate the morphology, mean interlayer spacing, crystallite size, porosity, and disordered carbon/graphitic carbon ratio. Finally, the carbons were used to prepare electrodes and their electrochemical properties were tested in LIB and Li-S batteries.

## 2. Materials and Methods

### 2.1. Preparation of the Activated Carbons

Four organic products of vegetable origin have been used as precursors for the preparation of hard carbons: walnut shells, apricot kernels, and cherry and olive stones. The carbonization procedure was carried out via pyrolysis of a small amount of the woody precursors (about 5–10 g) that were previously cleaned and chopped. The organic samples were placed inside a quartz tube and heated up to a temperature of 825 °C, with a heating rate of 5 °C min^−1^. Once this temperature was reached, it was maintained for 2 h. The pyrolysis was carried out in argon fluxed at 30 mL min^−1^. The so-obtained carbon was milled for 15 min, treated with 50 mL of 0.1 M HCl, and filtered. The residue was rinsed until the washing water was neutral, then 50 g of a solution of KOH 7 M was added. The suspension was stirred for 2 h. After two days of rest, the carbon was filtered without washing. This treatment left the carbon impregnated with the alkaline hydroxide. The KOH-impregnated carbon was heat-treated at 825 °C for 2 h in an argon flow (30 mL min^−1^) at the same heating rate (5 °C min^−1^). After cooling, the sample was rinsed until the washing water was neutral. To eliminate the humidity traces, the carbon was heated at 120 °C for 24 h. The scheme in Figure 1 illustrates the various steps of the preparation procedure.

### 2.2. TG-DTG Analysis

A TA Instruments Q600 system analyzer was used to record simultaneous TG-DTG curves in nitrogen gas at a flow rate of 100 mL min^−1^. Aluminum oxide was used as the reference material. To calibrate the temperature, the nickel Curie point was used as a reference. The weight of the samples was about 10 mg: the samples were placed in an alumina crucible and heated from room temperature up to 950 °C at a heating rate of 10 °C min^−1^.

### 2.3. X-ray Analysis

To evaluate the structure and other parameters (interlayer distance of graphene sheets (*d*_002_) and average size of graphitic domain (*L_C_*)), a Smart Lab Rigaku diffractometer equipped with CuKα radiation was used. The values of the interlayer distance were calculated using Bragg’s law.

### 2.4. SEM Characterization

All the samples were morphologically examined via scanning electron microscopy using a Tescan Vega 3 with LaB_6_ source. High-magnification photomicrographs were carried out with an AURIGA dual column focused ion beam, CrossBeam Workstation.

### 2.5. Raman Analysis

Raman measurements were performed using a home-built apparatus equipped with a Horiba-Jobin-Yvon spectrometer composed of a liquid-nitrogen-cooled CCD detector coupled to a TRIAX 550 monochromator with interchangeable gratings. The entrance slit of the monochromator is fiber-coupled to the optical sampling head based on a metallographic microscope with interchangeable objectives. Laser excitation is provided by a frequency-doubled Nd:YAG laser operating at 532 nm and spectra are collected in backscattering configuration; rejection of elastically scattered light is obtained by an edge filter. Spectra were measured on pressed carbon powders using a 1200 grooves/mm grating and 60 s acquisition time with a 32× Olympus objective, employing typical powers on the sample in the range of few milliwatts to avoid sample heating, and measurements were repeated 4 times on different sample positions. Accurate calibration of the measured Raman shift was obtained using the lines of a cyclohexane sample as references. Data were analyzed in the 1000–2000 cm^−1^ region using a multi-peak fitting software (Origin 2015, OriginLab Corp. Northampton, MA, USA): after subtraction of a smooth background and normalization to 1 using the intense Raman peak at 1340 cm^−1^, a nonlinear least-squares fitting procedure was used to separate the different spectral contributions and obtain their respective center positions, widths, and integrated areas, assuming Lorentzian shape; the fitting procedure was also used to obtain the uncertainty of the fitted bands parameters. Derived parameters used in the analysis discussed below and the associated uncertainties were calculated by averaging the values obtained from the measurements at different sample points.

### 2.6. Electrochemical Characterization

The electrodes were prepared mixing the hard carbon with poly-tetrafluoroethylene (PTFE) for at least 10 min in an agate mortar. The percentage of carbon was 85 wt.%. The plastic-like material so obtained was rolled between two stainless steel rollers to reduce the thickness to about 70 μm. Electrodes were punched with a diameter of 10 mm. The electrochemical performance was evaluated in a two-electrode lithium cell, in which the lithium acted both as working and reference electrode. A glass fiber was used as separator. The electrochemical performance was evaluated in 2032-type coin cells. A 1.0 M solution of lithium bis-trifluoromethanesulfonylimide (LiTFSI) in a 1:1 mixture of DOL/DME was used as electrolyte. When used as the cathode, 0.1 M LiNO_3_ and a 0.1 M polysulphide mixture were added to the electrolytic solution. The lithium polysulfide solution (Li_2_S_8_) was prepared by mixing Li_2_S and elemental sulfur in the molar ratio of 1:7 in tetraethylene glycol dimethyl ether (TEGDME). The mixture thus obtained was heated to 80 °C under magnetic stirring for at least 48 h, giving rise to a dark brown solution. The final concentration of Li_2_S_8_ was estimated to be 0.5 M. The cycling tests were automatically carried out with a battery cycler (Maccor 4000). All experimental activities were performed at 20 °C in a dry room (R.H. < 0.1% at 20 °C).

## 3. Results

### 3.1. TG-DTG Analysis

The thermogravimetric (TG) analysis was carried out to identify the pyrolysis process and behavior during the heating treatment in the presence of KOH in an inert atmosphere. All the samples showed a similar behavior, ascribable to a multistep pyrolysis process. Differences between the various precursors have been observed in the total weight loss and reaction temperatures in the TG and in the amplitude and shape of the peaks in the derivative thermogravimetric (DTG) curve. As an example, the TG and DTG of the apricot kernels and cherry stones hard-carbon-impregnated with KOH are shown in Figure 2. 

For both the samples, a first significant weight loss started at about 100 °C and ended at about 200 °C. This loss can be associated with the physical desorption of water or small molecules, such as CO_2_, coming from KHCO_3_ decomposition [21]. The weight loss occurred in two stages corresponding to two exothermic peaks in the DTG curve. For the apricot kernel, a further mass loss was observed over 200 °C, which was characterized by two significant mass reduction steps, as well evinced by the derivative curve of the thermogravimetric profile. On the other hand, a weight loss stage in the 210–450 °C temperature range was observed for the cherry stone. These weight losses can be attributed to the releasing of H_2_O as a consequence of the calcination of the alkaline hydroxide and the formation of K_2_O. At this stage the reaction can be described as indicated in Equation (1):2MOH → M_2_O + H_2_O(1)

Rodenas et al. [22] have shown that when using KOH as activator, the development of H_2_ starts at around 400 °C and proceeds according to Reaction 2.
6KOH + 2C → 2K + 3H_2_ + 2K_2_CO_3_(2)

Basically, the reactions occurring in this temperature range determine the burning of carbon and the formation of pores. The free energy (ΔG) of Reaction 2 is positive at room temperature but it changes sign at about 570 °C. Due to the dilution with the inert gas, the hydrogen partial pressure is less than 1 atm and the temperature at which the process becomes spontaneous should be lower than 570 °C. Furthermore, the reactivity of the precursor would have a decisive effect on the activation process. Generally reactive (slightly organized) precursors should give rise to greater porosity with NaOH than with KOH; conversely, poorly reactive (well organized) precursors should form larger pores when treated with KOH [23]. In our case, the reaction described in Equation (2) for the cherry stone sample was completed at 450 °C, indicating that this precursor has an ordered structure. The activation process for the apricot kernel occurred at much lower temperatures, suggesting that this substrate is a reactive active precursor. By further increasing the temperature, a continuous loss of weight occurred up to 800 °C. This weight loss was due to the decomposition reaction of K_2_CO_2_, which implies the release of CO_2_. The DTG curve evidenced that the carbonate formed in Reaction 2 decomposed over 700 °C (Equation (3)) and totally disappeared at 800 °C, following this reaction:K_2_CO_3_ → K_2_O + CO_2_(3)

### 3.2. X-ray Analysis

XRD analysis was carried out on the different types of carbons before and after the activation process. In all XRD patterns, two broad peaks could be observed at 22° and 43°, which are assigned to the crystallographic planes of [002] and [101] in the disordered and amorphous carbon structure [24]. The position of the peaks remained almost constant, except for small variations, even by changing the type of precursor. On the contrary, passing from one precursor to another, a different amplitude of the peaks was observed. Another variation was observed at the [002] plane before and after activation. To illustrate this effect, the diffractograms for the sample coming from the apricot kernel before and after activation with KOH is reported in Figure 3. 

The widening of the two peaks indicates a very low degree of graphitization and the presence of an extensive amorphous phase. While the position of the peak at 43.4° remained constant after the treatment with KOH, there was a small variation of the peak at 22°, which reaches 22.2°. The shift of the [002] peak to higher angles indicates that the thermal treatment with KOH decreased the interlayer distance between the graphitic sheets. Depending on the values of the interlayer distance, hard carbon microstructures can be classified as highly disordered, pseudo-graphite, and graphite-like. The first group includes distinguished hard carbon with a *d*-spacing larger than 0.40 nm, containing highly disordered micro crystallites [25]. Pseudo-graphite hard carbons generally show a *d*-spacing between 0.35–0.4 nm [26]. Finally, graphite-like microstructures have an interlayer distance smaller than the 0.35 nm. The hard carbons we obtained had a d-spacing equal or larger than 0.40 nm, so they were classified as highly disordered carbons. This kind of carbon has been found to be particularly adaptable to use as an anode in Na-ion batteries since the large interlayer spacing is good for transferring sodium ions [27]. The values of the interlayer distance can be calculated by rearranging the Bragg’s law: *d* = *λ/2 sin*(*θ*), where *d* is the distance between the adjacent sheets or layers, *λ* is the wavelength of the X-ray beam (1.5418 Å), and *θ* is the diffraction angle. As can be seen in Figure 3, the diffraction peak positions for the [002] plane were 2*θ* = 22.0° and 22.2° for the precursor and the activated carbon, respectively. Thus, the value of the interlayer distance decreased from *d*_002_ = 4.04 Å for the starting carbon to *d*_002_ = 4.00 Å for the activated one. A similar peak position shift to high angles was observed in carbonized corn straw piths with an increase in the reaction temperature [24]. The thickness and average width of the graphitic domains, *Lc* and *La*, can be calculated by using the following formula *L* = *λ*/*β cos*(*θ*)*,* where *β* is the full-width half-maximum, *λ* is the wavelength of the X-ray beam, and *θ* is the diffraction angle (22.2° for *Lc* and 43.4° for *La*). The *Lc* and *La* were 0.74 and 0.90 nm, respectively. By dividing the average thickness of the graphitic domains for the typical graphite interlayer distance (0.38 nm) [28], it was possible to calculate that they were composed of 1.9–2.4 stacked graphene layers.

### 3.3. SEM Characterization

Figure 4 shows the SEM images obtained at high magnification of the hard carbon, prepared using the apricot kernel before and after activation with KOH. The surface of the carbon appears to have been formed by graphene sheets rolled up one on top of the other. The graphene sheets of the sample that only underwent pyrolysis had a very smooth surface crossed by a series of streaks, all oriented in the same direction. After activation with KOH, the sample showed a structure that closely resembled that of the precursor. The same trend was observed for all the other samples for which the materials before and after the KOH treatment had a similar morphology.

Figure 5 shows the morphology of the hard carbons at low magnification after activation with KOH. All carbons appeared as aggregations of particles, ranging in size from a few microns up to 20 μm. The carbon from the apricot kernel had aggregates of different shapes and sizes. On the other hand, the carbon from the cherry stone looked more uniform. Carbon from the olive stone appeared as a flat surface, on which formations of different sizes emerged. The carbon of the walnut shell resembled the carbon of the apricot kernel since it was inhomogeneous with grains of different sizes. At higher magnification (Figure 6), all the carbons showed a flat surface of different extensions covered with small particles. Among the others, the carbon coming from apricot kernel and olive stone had particles with larger surfaces. Several voids were observed on the carbon surface, especially for carbon from the olive stone. Carbon from the cherry stone and walnut shell presented lower extensions of the flat surfaces, but while the cherry stone was rich in rounded small particles, the walnut shell appeared to be formed only by particles with a flat surface.

### 3.4. BET Measurements

The BET measurements showed small differences in the surface area of the various KOH-activated carbons. The surface area was observed to vary from 317 m^2^g ^−1^ for the carbon coming from walnut shell up to 347 m^2^g ^−1^ for the carbon coming from cherry stone.

### 3.5. Raman Analysis

The most significant Raman spectral region for carbon materials [29] is in the range 1100–2000 cm^−1^, where two main bands are apparent, namely the so-called G band around 1580 cm^−1^, which is narrow (less than 20 cm^−1^) in samples formed by good-quality graphite networks, and is assigned to in-plane sp2 carbon stretching vibrations, and the D (or D1) band around 1350 cm^−1^, assigned to the breathing mode of sixfold aromatic rings, which is forbidden in perfect graphite, but becomes active in presence of disorders, defects, or heteroatomic impurities. Other minor intensity bands appearing in defective samples, whose attribution is somewhat still debated [30], are located around 1150 (D4 band), 1500 (D3 band), and 1620 (D2 band) cm^−1^. The 1150 cm^−1^ peak (D4) appears only in very highly defective materials and is attributed to hydrocarbon components or aliphatic moieties grafted on to the basic structural units. The 1500 cm^−1^ component (D3) is a wide band attributed to defects outside the plane of aromatic layers, like tetrahedral carbons. Lastly, the 1620 cm^−1^ band (D2) appears sometimes as a shoulder on the G band and is attributed to graphene layers at the surface of a graphitic domain. In the samples under analysis, we found that experimental spectra could be satisfactorily fitted by a combination of the D1, G, D3, and D4 bands; in fact, it was not necessary to include the D2 band contribution. 

A representative example of experimental data with their spectral decomposition is shown in Figure 7. Information on the samples structure and composition can be obtained from the relative intensities (areas), widths, and spectral positions of the bands forming the Raman spectra. For all the samples analyzed in this work, we did not notice significant spectral variations as a function of the sample measured point; moreover, all spectra showed that the D1 disorder band was the most intense Raman component, with a G band width much larger (about 70 cm^−1^) than that observed in good quality graphitic samples, indicating a highly defective graphite network. Given the presence of four bands, each described by three parameters (position, width, area), it is useful to introduce a limited number of derived parameters to allow comparison of the sample quality.

Following the literature [31], here we used the integrated band intensities (*I*) to calculate the parameter *R*1 = *I_G_*/*I_D_*_1_, indicating the degree of graphitization; parameter *R*2 = *I_D_*_1_/(*I_G_* + *I_D_*_1_) indicating the degree of structural organization; parameter *R*3 = *I_D_*_3_/*I_G_,* indicating the amount of amorphous carbon; and the integrated intensity of the D4 band *I_D_*_4_, indicating the content of hydrocarbons or aliphatic compounds. As anticipated in the Materials and Methods section, *R*1, *R*2, *R*3, *I_D_*_4_, and the associated uncertainties were calculated from the average values and uncertainties of the band parameters obtained using the multi-peak fitting procedure on the data measured from different points on the samples. Results obtained for the samples used in this study are shown in Figure 8. 

Inspection of this Figure revealed that *R*1 (indicating the graphitization degree) showed a maximum for the cherry stones and a minimum for walnut shells. On the other hand, *R*2 (indicating the degree of structural order) did not show a great dependence on the carbon source, with its variation limited to absolute values and, in comparison, to the associated uncertainties. Differently, the amount of amorphous carbon (parameter *R*3) seemed to have a maximum for carbons obtained from apricot kernel and a minimum for carbons coming from walnut shells. Moreover, the content of hydrocarbons or aliphatic moieties (D4 band area) seemed to have a maximum for carbons coming from cherry stones. All in all, carbons obtained from walnut shells seemed to have the lower content of amorphous and other non-sp^2^ carbon.

### 3.6. Electrochemical Characterization

All samples were analyzed from an electrochemical point of view as cathodes for Li-S batteries or anodes for sulfur/LIB. In the first case, the cell was cycled between 0.05 and 1.00 V. In the second case, the voltage was stopped at 1.70 V upon discharge and then raised up to 2.80 V in charge. The carbonaceous matrices based on the walnut shell and cherry stone, both activated with KOH, gave the best electrochemical results when used both as anodes and cathodes in Li-S cell. Figure 9 shows the voltage profiles (Figure 9a,c) and the variation of the specific capacity (Figure 9b,d) as a function of the number of cycles for the walnut shell and cherry stone, respectively, when used as the anode in LIB. 

The electrodes exhibited a similar behavior. During the first discharge cycle, a voltage plateau around 1.2–1.0 V was observed, then the cell voltage began to lower progressively. When the cell voltage reached 0.20 V, a further variation in the curve slope tended to flatten the cell voltage. On the other hand, profound differences were observed in the specific capacity values. Table 1 reports the first-cycle specific capacity in discharge (calculated at different cell voltage values) and in charge, and the ratio between the two values. 

The electrode prepared with walnut shell carbon showed specific capacities approximately 2.78–2.45 times higher than the other electrode. This means that the voltage profiles of the two types of carbon were perfectly super imposable, differing only in the values of the specific capacity. For both the electrodes, a large first-cycle irreversible capacity was observed. The ratio between the first charge and discharge capacities were 0.28 and 0.31 for the walnut shell and cherry stone carbon-based electrodes, respectively. This irreversible capacity is detrimental for proper use as the anode in LIB. In the following cycles a slight capacity fade was observed for both the electrode. After 30 cycles, the charge specific capacity was reduced to 122 and 80 mAhg^−1^ for the walnut shell and cherry stone carbon-based electrodes, respectively. 

Figure 10 shows the voltage profiles (Figure 10a,c) and the variation of the specific capacity (Figure 10b,d) as a function of the number of cycles for the walnut shell and cherry stone, respectively when used as cathodes in Li-S batteries.

Both the electrodes were characterized by the typical charging plateaus of Li-S batteries. The two distinct plateaus were centered at 2.4 and 2.0 V in discharge, while during the charge the plateaus moved up to 2.2 and 2.4 V, corresponding to the oxidation of Li_2_S_4_, the main redox intermediate present in the TEGDME solution. The electrode prepared with the carbon obtained from the pyrolysis of walnut shells showed an initial specific capacity at the first cycle of about 1600 mAhg^−1^. In the next charge, the electrode was capable of recharging only 1500 mAhg^−1^. Capacity rapidly decreased to 1250 mAhg^−1^ during the second discharge and then dropped to 1150 mAhg^−1^ in the next cycle. In subsequent cycles, there was a slow decline in performance and the specific capacity reached 1000 mAhg^−1^ at the 60th cycle. During the cycles, part of the charged capacity was not recovered in the next discharge, so that the Coulombic efficiency was 91% (top right). Cherry stone carbon-based electrodes showed an initial first-cycle specific capacity of nearly 1800 mAhg^−1^, which decreased to 1180 mAhg^−1^ at the second cycle. A further significant decrease in the specific capacity was observed at the third cycle and after that the capacity remained stable over the next 60 cycles. In addition, in this case, the amount of accumulated charge was greater than that discharged, but the Coulombic efficiency was better than that exhibited by the other electrode, settling at 95% (bottom right). The origin of this low Coulombic efficiency can be ascribed to the polysulfides present in the electrolyte. Polysulfides can be reduced on the lithium surface to form short-chain polysulfides that can diffuse back to the cathode, where long-chain polysulfides are formed again. This process reduces the Coulombic efficiency and can lead to lithium corrosion. To reduce this process, LiNO_3_ was added as an additive. LiNO_3_ forms a passivation layer on the surface of the Li electrodes. This passivation layer remarkably diminishes the reduction of polysulfide species by the reactive lithium anode, increasing the Coulombic efficiency. This passivation layer, during the cycling process, can break and must be reformed, consuming charge, which is reflected in sudden changes in the Coulombic efficiency. To compare our results with those reported in the literature, we can refer to a recent review reporting the key aspects of biomass-derived activated carbon as a sulfur host and an analysis of the latest advances in its application in Li-S batteries [32]. The first cycle capacities recorded using walnut shell and cherry stone carbon are the best among those exhibited by carbon or activated carbon from agri-food residues when used as cathodes in Li-S cells. However, it must be considered that our system differs from the classic Li-S cells as the sulfur was introduced by adding a polysulphide solution to the electrolyte.

It is difficult to find a correlation between the carbon characteristics, as evidenced by chemical-physical measurements, and their electrochemical behaviors. Firstly, because both the morphologies, as evidenced by the SEM measurements, and the crystal structures, as they emerged from the X-ray analysis, were very similar between the different carbons. Consequently, it is not possible to attribute the different electrochemical behavior exhibited by the electrodes to differences in the structure or shape of the materials. Major differences were highlighted by Raman spectroscopy. In particular, the carbons that showed the best cathode characteristics in Li-S batteries had the lowest degree of graphitization and the lowest percentage of other non-sp^2^ carbons, and the highest degree of graphitization and the highest percentage of other non-sp^2^ carbons. Therefore, these two parameters must not be related to the electrochemical properties. These carbons had similar average values of the degree of amorphization, which therefore could be correlated to the electrochemical activity. However, this does not explain why the carbon coming from the olive stone, which had a similar degree of amorphization, did not have the same electrochemical properties of the other two carbons (data not shown). The carbon coming from the cherry stone showed the best characteristics as an anode in LIB, i.e., it had the highest degree of graphitization and the highest percentage of other non-sp^2^ carbons, characteristics that could be associated with its good electrochemical performance. These speculations are not very effective, and it is evident that other characteristics of the carbons must be identified in order to be able to correlate them with their electrochemical activity.

## 4. Conclusions

Four different hard carbons have been prepared from waste of vegetable origin. These carbons had a highly disordered structures and morphologies characterized by the presence of graphenic planes superimposed on each other. The distances between these graphene planes were very large when compared to graphite. The carbons were activated by a heat treatment in the presence of KOH. Activated hard carbons from cherry stones and walnut shells have proven to be particularly suitable for use as an active material for making cathodes for Li-S batteries. Additionally, activated hard carbon coming from a cherry stone also exhibited good activity when used as an active material in anodes for LIB. Although Raman spectroscopy showed some differences in the degree of graphitization, in the percentage of amorphous carbon, and in the presence of other types of non-sp^2^ carbon, it was not possible to correlate these characteristics with the electrochemical performances exhibited by the hard carbons.

## Figures and Tables

**Figure 1 nanomaterials-12-01349-f001:**
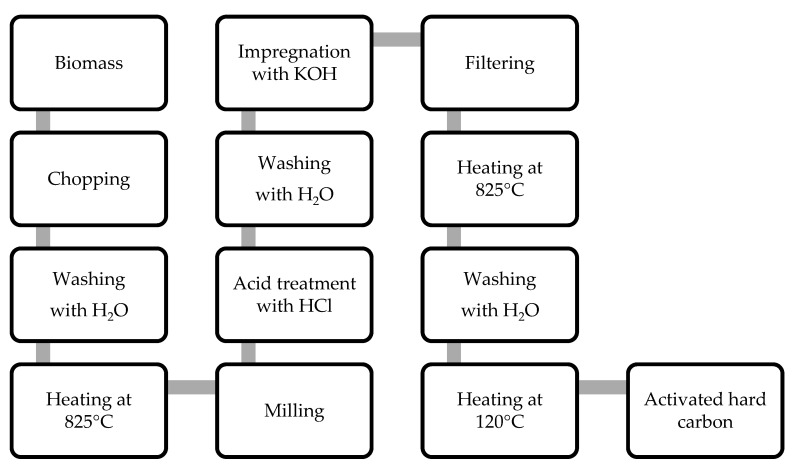
Schematic diagram illustrating the different phases of the production process of the activated hard carbon starting from biomass.

**Figure 2 nanomaterials-12-01349-f002:**
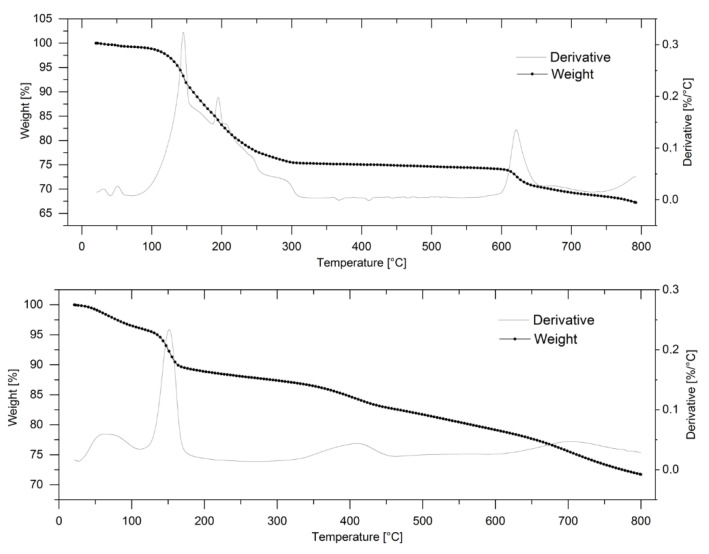
TG and DTG analysis of carbons coming from apricot kernel (**upper panel**) and cherry stone (**lower panel**) added with KOH. Heating rate 10 °C min^−1^ in nitrogen flow.

**Figure 3 nanomaterials-12-01349-f003:**
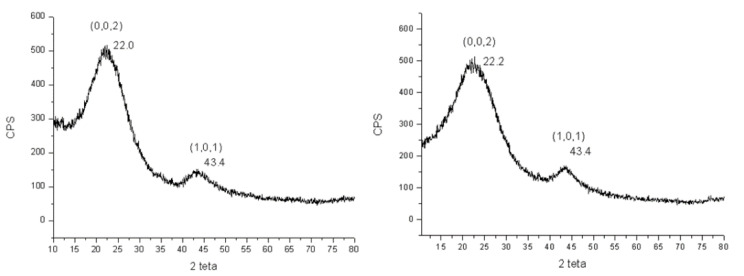
Diffraction peaks observed for the apricot kernel carbon before (**left**) and after (**right**) activation with KOH.

**Figure 4 nanomaterials-12-01349-f004:**
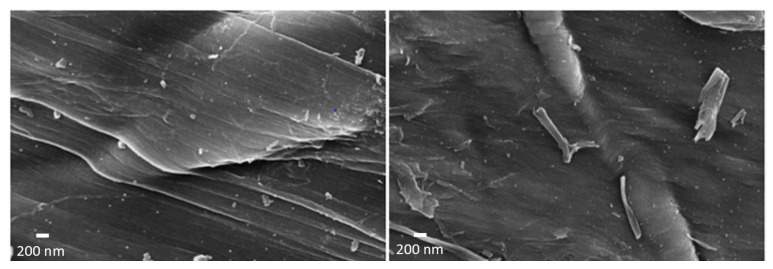
On the left is the hard carbon coming from the apricot kernel before the activation with KOH. On the right, the same carbon is shown after the activation treatment.

**Figure 5 nanomaterials-12-01349-f005:**
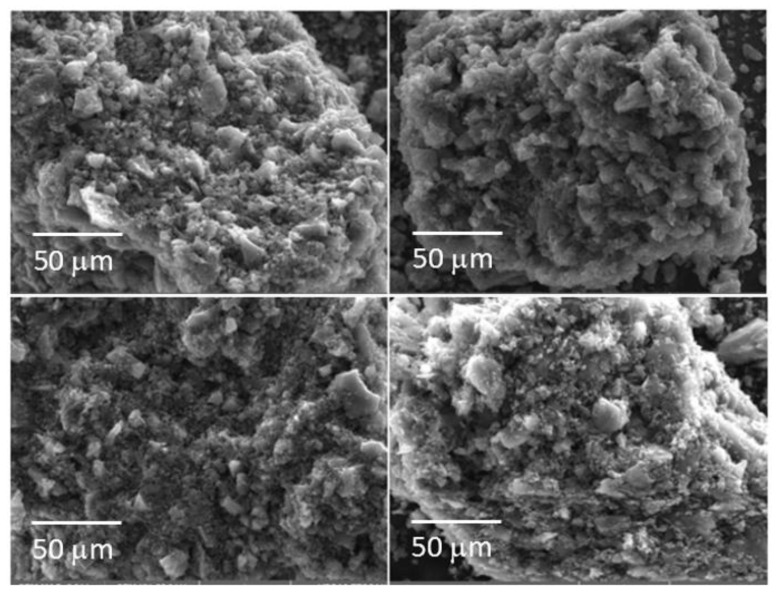
Morphology of the hard carbons after activation with KOH recorded at low magnification. (**Upper left**): apricot kernel; (**upper right**): cherry stone. (**Lower left**): olive stone; (**lower right**): walnut shell.

**Figure 6 nanomaterials-12-01349-f006:**
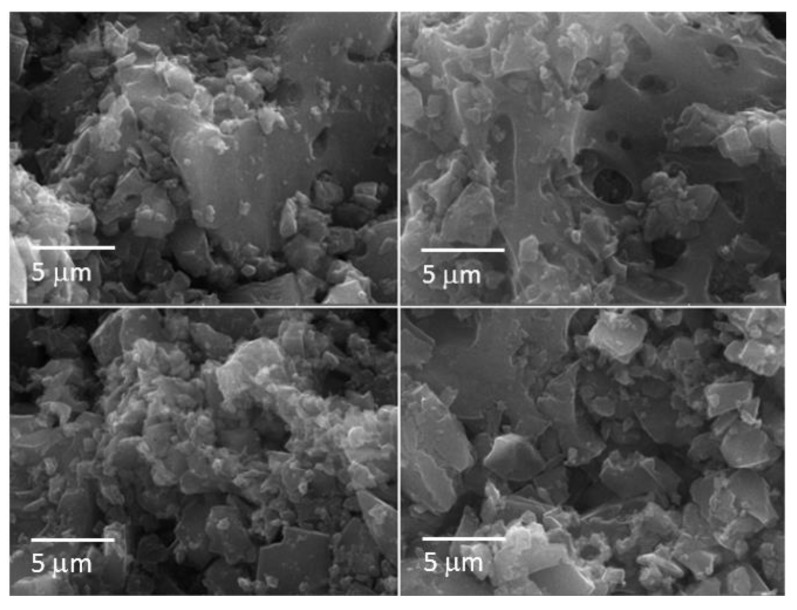
Morphology of the hard carbons after activation with KOH recorded at high magnification. (**Upper left**): apricot kernel, (**upper right**): cherry stone. (**Lower left**): olive stone, (**lower right**): walnut shell.

**Figure 7 nanomaterials-12-01349-f007:**
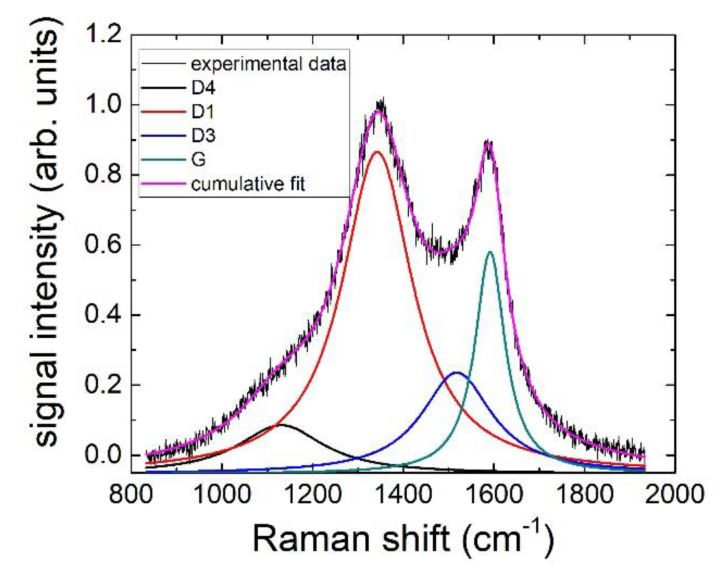
Raman spectrum measured on carbon obtained from the activated apricot kernel, together with its spectral decomposition into the D1, D3, D4, and G bands.

**Figure 8 nanomaterials-12-01349-f008:**
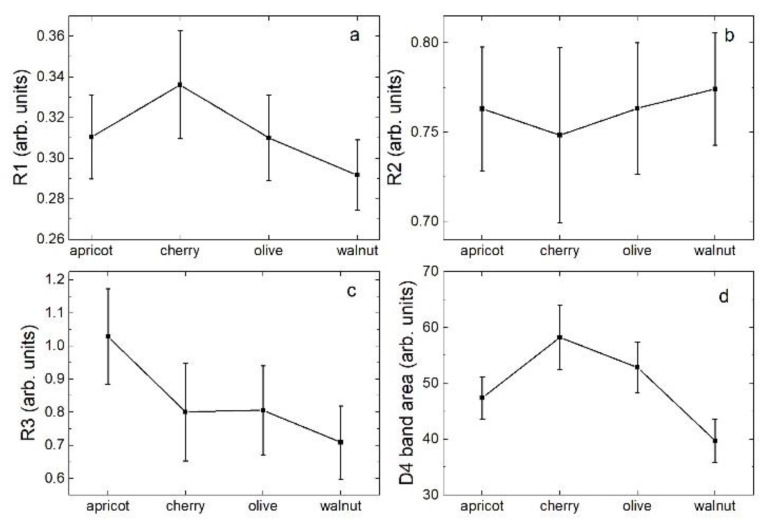
Values of the R1 (**a**), R2 (**b**), and R3 (**c**) parameters, and of the D4 band area (**d**) used to characterize the samples Raman spectra as a function of the source of the carbon powder as explained in the text.

**Figure 9 nanomaterials-12-01349-f009:**
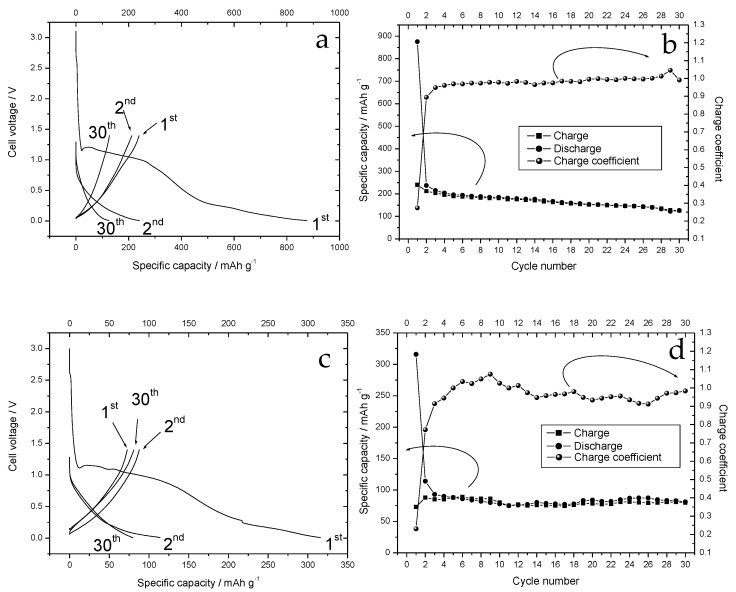
(**a**) Voltage profiles of the walnut shell carbon-based electrodes, during the first 30 cycles conducted at constant current. (**b**) Variation of the specific capacity and charge coefficient as a function of the number of cycles. The electrode load was 7.0 mg. (**c**) Voltage profiles of the cherry stone carbon-based electrodes. (**d**) Variation of the specific capacity and charge coefficient as a function of the number of cycles. Lithium metal was used as counter and reference electrode. Cycling was carried out at constant current between 0.05 and 1.00 V. The electrode load was 7.7 mg.

**Figure 10 nanomaterials-12-01349-f010:**
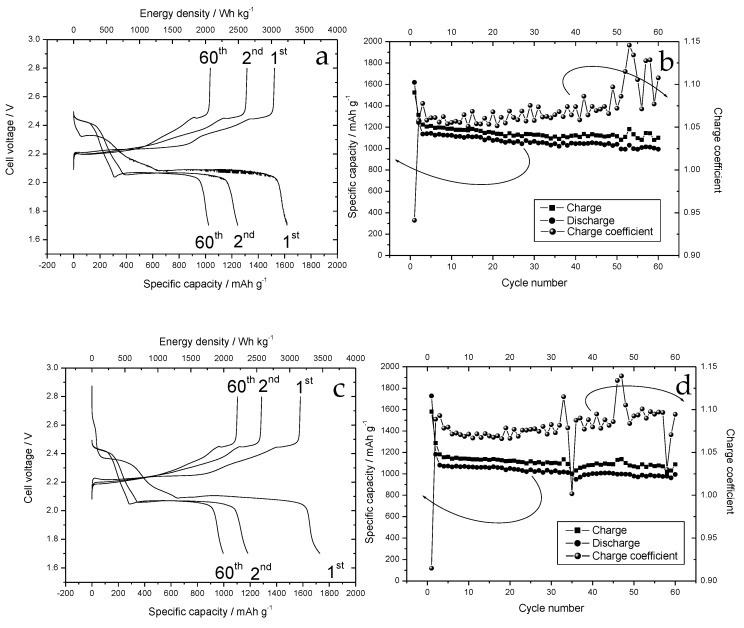
(**a**) Voltage profiles of the walnut shell carbon-based electrodes, during the first 60 cycles conducted at a constant current. (**b**) Variation of the specific capacity, energy density, and charge coefficient as a function of the number of cycles. The electrode load was 4.5 mg. (**c**) Voltage profiles of the cherry stone carbon-based electrodes. (**d**) Variation of the specific capacity, energy density, and charge coefficient as a function of the number of cycles. Lithium metal was used as counter and reference electrode. The electrode load was 4.1 mg. The specific capacity and the energy density were calculated on the weight of the polysulfide in the electrolyte. Cycling was carried out at a constant current between 1.70 and 2.80 V.

**Table 1 nanomaterials-12-01349-t001:** First cycle specific capacity in discharge (calculated at different cell voltage values) and in charge. The first two rows report the specific capacity. The last row reports the ratio between the specific capacities.

Specific Capacity mAhg^−1^	Down to 1.00 V	Down to 0.20 V	Down to 0.05 V	Up to 1.00 V
Walnut shell	250	600	850	240
Cherry stone	90	230	320	98
Ratio	2.78	2.73	2.66	2.45

## Data Availability

Not applicable.

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
