# Peer review of "Hard Carbons for Use as Electrodes in Li-S and Li-ion Batteries"

_nanomaterials, 2022, doi:10.3390/nano12081349_

Round 1

Reviewer 1 Report

The paper is well structured and clear. They are interesting results although it is more of empirical nature, except the Raman data that give interesting insights. The scientific results are a bit disappointing. There are no errors given in the electrochemical data, so are the tests done only on one cell? It would be good to mention how many testscells were prepared so if the data are reproducible.

The XRD data are missing an explanation why the interlayer distance of the samples is so high. With 4 Angstöm compared to 3.34 Angström I wonder is this really graphite?

In Figure 2, the 101 reflection has the same value. If the 002 reflection would be shifted ti 4.04 Angström as claimed, this should also be visible in the 101 reflection as the 00l is part of this. For graphite, as there are only few reflections with preferred orientation, an internal standard would have been better to make sure that the positons are accurate.  It rather looks like the peak is broader for the KOH treated sample. This means that the grainsizes became smaller or have a larger grainzise distribution. It is possible to determine the domainsize in c-direction from XRD and in a-Direction (La) from Raman. So maybe you should at least mention why you did not look at these data.

A bit more literature on this can be found here

https://www.sciencedirect.com/science/article/abs/pii/S000862231731179X

I would appreciate to know why no higher temperatures for graphitisation were tested.

Some minor findings:

93 washed

94 there is a space missing before 2.2

172 further is the observed

210 Figure 2 – the cps on the y axis of the right side figure cannot be seen completely

Reviewer 2 Report

Pozio et al studied hard carbons used as electrodes in Li-ion and Li-S batteries. The study is interesting and can be considered for acceptance after a major revision.

1. Title. In this paper, the application of carbon materials in lithium-ion battery and lithium-sulfur battery is studied, but the title only describes lithium-sulfur battery. Therefore, it is suggested that the author modify the title to make the title correspond to the text.

2. The preparation process of the material includes several steps. It is suggested that the author give a schematic diagram of the preparation process of the material so that the reader can understand each step more clearly.

3. The author fabricated carbon materials from four vegetable origin, but did not give all the test results of the four materials. This prevents the comparison of the four materials from being clearly shown. Some tests can provide representative results, but there are some test results that must be provided. For example, the morphology comparison of SEM images of the four materials at low and high magnifications, as well as the cycling performance of lithium ion batteries and lithium sulfur batteries.

4. Please provide the following test results for at least one material. TEM images at different magnifications, SAED, BET, and CV of lithium-ion and lithium-sulfur batteries.

5. The authors should specify the problem to be addressed in the paper in the introduction and abstract part. 

6. The authors should compare the electrochemical properties of current and published works in a table.

7. The format of the references needs to be checked. In addition, some references are too old. Cite the recent work of carbon-related materials in energy storage fields. The following work is for reference:
[1]  Journal of Colloid and Interface Science 2020, 569, 22–33. DOI: 10.1016/j.jcis.2020.02.062.
[2]  JOURNAL OF ENERGY CHEMISTRY 2022, 64,  520-530. DOI: 10.1016/j.jechem.2021.05.005.
[3]  Journal of Alloys and Compounds 2022, 900, 163420. DOI: 10.1016/j.jallcom.2021.163420.
[4]  Nanomaterials 2020, 10, 346. doi:10.3390/nano10020346.

Reviewer 3 Report

Recommendation: not suitable for Nanomaterials.

Comments: In this work, the authors reported a series of biomass-derived hard carbons being used as electrodes in Li-S batteries. The morphology and electrochemical performance are poor and the manuscript and pictures are badly organized. Furthermore, there lack of mechanism investigation and reasonable structure evolution explain in the manuscript. The experiment data relevant to the biomass-derived hard carbon S  composite cathodes offered in this manuscript are insufficient to support the conclusion now. So, I think this manuscript is not suitable for Nanomaterials.

  1. The introduction of this paper needs to make a strong argument about the impact and novelty of the work further. So, the introduction should be enriched and some related biomass-derived carbon materials applied in Li-S batteriesshould be added in this section.
  2. To understand the difference between the walnut shells, apricot kernel, and cherry and olive stone-derived hard carbons, the SEM or TEM results of these carbon materials should be for comparison.
  3. Thecycling performance test must prolong. Why the coulombic efficiency is not stable and not close to 100%?
  4. The active mass loading of the electrodeand the total energy density should be also taken into consideration.
  5. The structure evolution during the charge/discharge process should be identified by in-situ XRD or XPS or Raman.
  6. The authors should compare the electrochemical performance with reported biomass-based carbon-sulfurcathode
  7. There are grammatical issues and writing mistakes in the manuscript. The authors should carefully check and correct them. “at 120°C for 24 h.2.2. TG-DTA analysis.” line 94 page 2. “Figure 2” on Page 5

Reviewer 4 Report

I believe that the paper entitled "Hard Carbons to Be Used as Electrodes in Li-Sulphur Batteries" is suitable for publication in Nanomaterials. The authors need to consider the following points prior publication

1.The comparison of the samples performance with the literature is important to strengthen the novelty of the work. How do your results compete with other reported for Li-S batteries?

2. Can you please comment on the structural and morphological characteristics of the samples after the cycling process? Do these characteristics change over continuous cycles?

Round 2

Reviewer 2 Report

The authors have revised the paper as suggested, in which case this work can be considered for acceptance.

Author Response

Thank you  for your help - Best regard.

Reviewer 3 Report

There was no substantial improvement in the quality of the articles. Some problems remain unresolved.

Author Response

Dear reviewer,
I am sorry that we have not been able to convince you about the goodness of our work. I would like to point out that we have tried to answer your comments point by point. As regards the electrochemical characterization, compared to the first version we have updated the figures with the latest results and brought the number of cycles up to 30 for the anode and 60 for the cathode. In our opinion, although the number of cycles we are reporting is not high, it is still sufficient to understand the behavior of the cells. Regarding the morphological characterization, we implemented the work in accordance with your suggestions adding the SEM images of all carbon materials. Images, at low and large magnification, were inserted into the work and the result of the analysis was adequately commented in the text.
We were unable to monitor the evolution of the structure through in situ measurements. I understand that such a type of measurement would have greatly increased the quality of the manuscript but unfortunately, we are not equipped for this type of measurement. However, we appreciated your advice, taking inspiration from his observation for a future work.
In conclusion, I hope you can recommend the publication of the paper in this revised version.